# Research on Spatial Delineation Method of Urban-Rural Fringe Combining POI and Nighttime Light Data—Taking Wuhan City as an Example

**DOI:** 10.3390/ijerph20054395

**Published:** 2023-03-01

**Authors:** Jing Yu, Yingying Meng, Size Zhou, Huaiwen Zeng, Ming Li, Zhaoxia Chen, Yan Nie

**Affiliations:** 1Hubei Key Laboratory of Regional Development and Environmental Response, Hubei University, Wuhan 430061, China; 2Shenzhen Urban Space Planning and Architectural Design Co., Ltd., Shenzhen 518039, China; 3Hubei Provincial Key Laboratory for Geographical Process Analysis and Simulation, Central China Normal University, Wuhan 430062, China

**Keywords:** NPP and POI composite index, urban-rural fringe, breaking point analysis method, urban and rural spatial structure

## Abstract

The boundary delineation of the urban-rural fringe (URF) is the basic work of fine planning and governance of cities, which plays a positive role in promoting the process of global sustainable development and urban and rural integration. In the past, the delineation of URF had shortcomings such as a single selected data source, difficulty in obtaining data, and low spatial and temporal resolution. This study combines Point of Interest (POI) and Nighttime Light (NTL) data, proposes a new spatial recognition method of URF according to the characteristics of urban and rural spatial structure, and conducts empirical analysis with Wuhan as the research object, combining the information entropy of land use structure, NDVI, and population density data to verify and compare the delineation results and field verification was conducted for typical areas. The results show that (1) the fusion of POI and NTL can maximize the use of the characteristics of the differences in facility types, light intensity, and resolution between POI and NTL, compared with the urban-rural fringe boundary identified by POI, NTL or population density data alone, and it is more accurate and time-sensitive; (2) NPP and POI (fusion data of Suomi NPP-VIIRS and POI) can quantitatively identify potential central area and multi-layer structure of the city. It fluctuates between 0.2 and 0.6 in the urban core area of Wuhan and between 0.1 and 0.3 in the new town clusters, while in the URF and rural areas drops sharply to below 0.1; (3) the urban-rural fringe area of Wuhan covers a total area of 1482.35 km^2^, accounting for 17.30% of the total area of the city. Its land use types are mainly construction land, water area, and cultivated land, accounting for 40.75%, 30.03%, and 14.60% of the URF, respectively. Its NDVI and population density are at a medium level, with values of 1.630 and 2556.28 persons/km^2,^ respectively; (4) the double mutation law of NPP and POI in urban and rural space confirms that the URF exists objectively as a regional entity generated in the process of urban expansion, provides empirical support for the theory of urban and rural ternary structure, and has a positive reference value for the allocation of global infrastructure, industrial division, ecological function division, and other researches.

## 1. Introduction

Urbanization follows the principle of serving the cities from the countryside and has played a major role in promoting the economic “take-off” of the world, especially in developing countries. At the same time, it has also led to problems such as bloated cities, rural decay, and ecological degradation and fragmentation [1]. As the frontier of urbanization, the urban-rural fringe (URF) has sharp conflicts in land use and governance issues of public health, environmental sanitation, social poverty, and fairness. It is considered to be the third element in the urban and rural structure, and its spatial structure research and scope have been determined has been a worldwide problem [2]. The delineation of the spatial scope of the URF, as a leading link in the fine planning and governance of urban and rural areas, affects the effective use of spatial resources and the sustainable and healthy development of URF. Scientific selection of data sources is of great significance in improving the accuracy of the delineation results [3].

In the early stage, the spatial definition of the urban-rural fringe mostly selected indicators such as the distance from the urban core [4], the number and proportion of the population [5], commuting distance [6], etc. For example, Russwurm defined the annular area extending 6~10 km outward from the urban center as the URF [4]. Bryant determined the range of URF according to the proportion of the agricultural population and non-agricultural population [7]. Such methods are greatly limited by administrative boundaries, and the accuracy of the results is low; they are difficult to apply to cities of different development levels and regions. With the further development of informatization, big data such as land use [8], landscape [9], and population spatial distribution datasets [10] are widely used in spatial quantitative research. For example, based on the edge effect theory, Huang et al. updated the information entropy model of land use structure and studied the URF of the Guangzhou-Foshan metropolitan area, providing an effective method for identifying the URF based on remote sensing images and land use [8]. Long et al. analyzed the landscape pattern and its dynamic evolution of the urban fringe in Wuhan and pointed out that Wuhan presents a clear three-circle structure of built areas, urban fringe areas, and rural hinterland [9]. Although the above methods break through the limits of administrative boundaries, they have a low degree of interpretation of economic and social development [11].

The population density distribution data are popular geographical spatial distribution data in recent years, which are simulated and generated by a spatial algorithm, and the inversion results can show the differences in population distribution within the administrative region. It is an important data source for GDP prediction [12], disaster prevention and mitigation planning [13], and urban and rural development planning [14]. Among them, WorldPop has the highest overall accuracy compared with commonly used kilometer-level open source data such as LandScan, GPWv4, and China kilometer grid population distribution data set [15,16], but the above data are retrieved from population census data, their temporal resolution is relatively low, which can only show the spatial difference in population size, it is difficult for them to reflect the connotative characteristics of population urbanization such as population occupational structure and lifestyle. Thus they have a low degree of interpretation of the “transition” characteristics of the URF. At the same time, open-source data such as OSM were usually selected for the development of population density data, which are relatively low in accuracy, especially in the town street scale, and require secondary development for application. Therefore, population density distribution data are relatively not applicable in the study of urban-rural fringe.

Nighttime Light (NTL) images are highly objective and can accurately reflect human activities and regional development differences. They have been widely used in population and GDP estimation [17], urbanization dynamics estimation [18,19], and boundary extraction between built-up areas and urban agglomerations [20,21]. In recent years, a series of studies have shown that the spatial attenuation of luminous remote sensing presents an inverse “S” curve that is fast at first, then slow, and then fast. This gradient change law can be applied to the spatial range identification of URF [22]. By analyzing the spatial change characteristics of DMSP/OLS nighttime light, Zhao et al. built a combined value model of light brightness and light brightness fluctuation characteristics and identified the urban-rural fringe in the main urban area of Chongqing, with a resolution of 1000 m [23]. Zhao et al. used DMSP/OLS nighttime light data as the data source and used the K-means method to identify the URF of Beijing with an accuracy of 1 km, too [24]. Although Luojia 1-01 has the highest resolution of 130 m, its official website only provides data from 2018 to 2019. At present, most of the current nighttime light used to divide the urban-rural fringe are DMSP/OLS with a precision of 1000 m and NPP-VIIRS with a precision of 500 m. Li et al. also found that the “saturation“ and “overflow“ effects of nighttime light and the limitation of its resolution may cause the recognition results to be larger than the actual range when using nighttime remote sensing, which is currently only practiced in mega cities such as Chongqing and Beijing [24].

The density distribution of point of interest (POI) in electronic maps can reflect the intensity of human activities and the level of social and economic development from both macro and micro levels, and also has a significant correlation with the distribution of plants, natural landscapes, and their patterns [25]. It has been widely used in the analysis of urban functional structure [26], extraction of built-up areas [27], measurement of land use mixing degree [28], spatialization of population data [29], and other studies. Lei analyzed the relationship between the density value and the distance of spatial change of POI in the urban agglomeration, and identified the urban fringe of the Guangzhou-Foshan metropolitan area using the method of breakpoint analysis, confirmed the feasibility of POI used for delimitation of urban-rural fringe [30]. However, when Qi et al. used the industry density of POI to identify the URF in Beijing, they found that the POI cannot reflect the scale and development intensity of the industry, which may lead to incorrect classification in industrial parks and other places.

The research confirms that POI and Nighttime Light have a good spatial coupling relationship, showing a strong consistency in the research results of urban spatial structure [31]. The combination of them can break through the restrictions of administrative boundaries in traditional statistical data. Compared with land use, landscape, population density, and other data, it can not only ensure the interpretation of social economy but also ensure a high spatial and temporal resolution. It is an ideal data source for the division of the URF. According to the characteristic of the fusion value, which is high in cities and low in the countryside, most of the relevant studies use the dichotomy method to identify built-up areas and non-built-up areas. However, there is little analysis of its internal spatial variation, and few studies have extended it to the URF. The URF in this study specifically refers to the large contiguous transition zone from the urban core area to the countryside formed by the rapid flow of population, industry, land, and other factors between urban and rural areas due to the increased demand for space expansion in the process of rapid urbanization [32].

This study takes Wuhan City as the research area; through analyzing the spatial distribution relationship between NPP and POI and the urban-rural fringe, it aims to introduce POI and Nighttime Light fusion data into the spatial delineation of the URF to improve the accuracy and timeliness of it. This method is expected to be applied to the practice of land spatial planning, and the delineation results will help to comprehensively understand the urban and rural spatial structure and provide references for the fine governance of the URF around the world.

## 2. Overview of the Study Area and Data Processing

### 2.1. Overview of the Study Area

Located at 113°41′–115°05′ E, 22°29′–31°58′ N, Wuhan is the capital city of Hubei Province and the only sub-provincial city in central China (Figure 1) [33]. It consists of 13 districts with an area of 8569.55 km^2^. By the end of 2019, Wuhan’s GDP had exceeded 1.62 trillion yuan, with a permanent population of 11.21 million, and its urbanization rate reached 80.49%. With the advancement of urbanization, Wuhan has gradually built a multi center and cluster urban spatial pattern through “internal optimization and external expansion”. The development of urbanization space in Wuhan is unbalanced. The urbanization rates of the seven central urban areas are all over 90%, while the urbanization rates of the surrounding built areas is low, such as Caidian District and Huangpi District, with urbanization rates of 40.63% and 49.21%, respectively [34].

### 2.2. Data Source and Preprocessing

#### 2.2.1. Data and Sources

The data in this study are divided into three categories: Nighttime Light data (NTL), Point of Interest (POI) data, and various auxiliary data. The DNB band of the visible, infrared imaging radiometer sensor carried by the Suomi NPP satellite from the National Geophysical Data Center of the United States is selected for the NTL image (https://payneinstitute.mines.edu/eog/, accessed on 14 March 2021), and the imaging time is November 2020 with a resolution of 500 m, the unit is nanoWatts/cm^2^/sr, and the value range is 0~472.86. POI data are crawled by Ospider software according to the API interface provided by Baidu Maps (https://lbsyun.baidu.com/index.php?title=webapi, accessed on 10 April 2021). The Normalized Vegetation Index (NDVI) from 2010 to 2020 was synthesized by the Modis vegetation index product MOD13Q1 downloaded from NASA’s official website, with a resolution of 250 m and a value range of −1~1 (https://modis.gsfc.nasa.gov, accessed on 17 May 2021). Land use data come from the Third National Land Survey, and population data come from Wuhan Statistical Yearbook (2020). The resolution of the WorldPop dataset in 2020 is 100 m, and the unit is the number of people per pixel, downloaded from the official website of the project (https://www.worldpop.orgl, accessed on 20 June 2021).

#### 2.2.2. Data Preprocessing

Nighttime Light processing: the global monthly data light products selected in this study have eliminated cloud layer reflection, atmospheric refraction, aurora, lightning, and other stray light and calibrated the satellite radiation. The ENVI median filtering tool is still needed to remove the background noise.

POI processing: The original data of Baidu map POI have problems with data redundancy and complicated classification. In view of the research needs, this study has cleaned and reclassified the original data, removed the approximate value, duplicate value, and null value, removed the administrative landmark category, the entrance and exit category, and the gate address category of road and building ancillary information that has no physical significance. Five categories of POI, which are closely related to natural landscapes and urban and rural population activities, are selected as the research objects, namely cultural and sports, commercial, industrial, public service, and residential. Finally, 345,861 valid POI of Wuhan in April 2021 were finally obtained (Table 1).

Land use processing: Reclassify the data of the third national land survey into eight categories: cultivated land, garden land, forestland, grassland, industrial land, other construction land, waters, and others. Among them, the industrial land is separated from the construction land, which is the key land type indicating the urban-rural fringe, and follows the general change law of the rural-urban land circulation from suburban agricultural land → vegetable land → industrial land → residential land filling → commercial service facility land [35].

## 3. Research Design

### 3.1. Research Methods

#### 3.1.1. Kernel Density Estimate (KDE)

Kernel density analysis can calculate the density of various spatial elements in their surrounding neighborhood and continuously simulate and visualize the spatial distribution of various elements. This paper uses this method to visually express the relative concentration of Wuhan’s POI data. The calculation formula is as follows [32]:(1)$$P_{i}=\frac{1}{n\pi R^{2}}\times \sum_{j=1}^{n}K_{j}{\left(1-\frac{D_{ij}^{2}}{R^{2}}\right)}^{2}$$
where, $P_{i}$ is the calculated value of the kernel density of any *i* point in space; *R* is the search radius of the kernel density function; n is the total number of *j* points of the research object within the bandwidth *R*; $K_{j}$ is the weight of *j* point. *R* > $D_{ij}$ and $D_{ij}$ is the Euclidean distance between *i* and *j*.

#### 3.1.2. NPP and POI Composite Index

Based on the feature that the brightness value of night light data and the nuclear density value of POI data are positively correlated with urban development, the index adopts the geometric average method to fuse the DN value of nighttime light data and the nuclear density value of POI data, which can eliminate the huge difference between the two units of magnitude [11]. This method can not only retain the contiguous advantages of the recognition results using NTL but also reduce its “saturation” and “overflow” effects. The formula is as follows:(2)$${POINTL}_{i}=\sqrt{P_{i}\times {NPP}_{i}}$$
where ${POINTL}_{i}$ represents the NPP and POI composite index; $P_{i}$ is the POI core density value at point *i*; ${NPP}_{i}$ is the Nighttime Light brightness value of point *i*.

#### 3.1.3. Breaking Point Analysis

The breaking point analysis is a theory about the interaction between cities and regions proposed by P.D. Converse. It is widely used to determine the range of urban attraction by selecting the maximum value of distance attenuation as the mutation termination point [36]. The formulas are as follows:(3)$$V_{ij}=X_{i(j+1)}-X_{ij}$$
(4)$$\bar{V}={\frac{1}{n}\sum V}_{ij}$$
(5)$${DDV}_{ij}=\frac{V_{ij}}{\bar{v}}$$
(6)$${DDV}_{i}=max({DDV}_{ij})$$
where $V_{ij}$ is the change rate on the *j* section of the *i*-th section line; $X_{i(j+1)}$ is the *j* + 1 sequence characteristic value on the *i*-th section line; $X_{ij}$ is the *j* sequence characteristic value on the *i*-th section line; $\bar{V}$ is the average change rate on the *i*-th section line; ${\bar{D}DV}_{ij}$ is the distance change value on the *i*-th section line; ${DDV}_{i}$ is the maximum distance change value on the *i*-th section line, that is, the breakpoint in the *i*-direction.

#### 3.1.4. Land Use Structure Information Entropy Model

The information entropy of land structure can reflect the complexity of land use structure in a certain region. The higher the entropy value is, the more complex the land use structure is, and the more diverse the land types are, and vice versa [37]. The calculation formula is as follows:(7)$$W={\sum_{i=1}^{n}x}_{i}\ln x_{i}$$
where *W* is the information entropy of land use structure; *i* is the number of land types in the sample area; $x_{i}$ is the percentage of the total area of a certain land use type in the sample area. The larger the *W* value is, the more complex the land use structure in the sample area is, and on the contrary, the simpler it is.

### 3.2. Standards for Spatial Delineation of Urban-Rural Fringe

#### 3.2.1. NPP and POI Composite Index Construction

Based on the difference between POI and NTL formats, this study uses the kernel density analysis to convert POI point data into grid data that can intuitively represent the spatial distribution and uses the NPP and POI composite index and fuzzy membership tool to calculate the normalized NPP and POI comprehensive index, which is used for the spatial range delimitation of the URF. In the analysis of POI core density, the determination of bandwidth value has a great impact on the calculation results of the NPP and POI composite index [38]. For the nuclear density analysis of POI, empirical values are usually used for bandwidth settings in the academic community. For example, Lei used 3500 m as the POI core density analysis bandwidth to divide the urban fringe of the Guangzhou-Foshan metropolitan area [30]. Li et al. used 500 m as the bandwidth of POI nuclear density analysis to construct the NTL and POI composite index to extract the built-up area of Nanjing [11]. CL et al. used 50 m as the bandwidth of POI nuclear density analysis to develop an effective method for POI/ROI discovery from Flickr [39]. This study initially uses the most commonly used kernel density analysis bandwidth value of POI in the spatial delimitation of the URF, that is, 3500 m, to divide the URF in Wuhan. It is found that although the spatial continuity of the results is good, the degree of reduction of NPP-VIIRS “saturation” and “spillover” effects is low, and it is difficult to display the local details of urban and countryside. After that, this study combined the research experience of the application of POI in Nanjing, Wuhan, and other megacities, selected 50 m, 500 m, and 3500 m as the nuclear density analysis bandwidth values, respectively, and made a comparative analysis of the spatial morphology of NPP and POI (Figure 2).

It can be seen from Figure 2a that, affected by spatial resolution, NPP/VIIRS has a significantly higher brightness value in highly urbanized areas and spreads to adjacent spaces along the main traffic lines. The lights in Qianchuan Street in the north are even more tail-flicking. The boundary between urban and rural areas is unclear, and there are “saturation” and “overflow” phenomena. The fusion results of POI and NPP/VIIRS under different bandwidth analyses can improve this phenomenon to varying degrees. As can be seen from Figure 2b–d, when the bandwidth value is 50 m, NPP and POI presents a dotted broken distribution feature, and the data space continuity is poor. When the bandwidth value is 3500 m, the “saturation” and “overflow” effects of NPP/VIIRS are reduced to a low degree, and local details of urban and rural space are difficult to highlight. When the bandwidth value is 500 m, the “saturation” and “overflow” effects of NTL can be significantly reduced, it can also integrate the differences between facility types and light intensity, reduce the problem of strong light values along the road, and meet the needs of balancing the overall scale and local details in the delineation of urban-rural fringe. Compared with the recognition results of 3500 m, the overall distribution of the urban-rural fringe is the same, but the outer boundary is slightly inward. Therefore, 500 m is determined as the POI core density analysis bandwidth value in this paper to generate the NPP and POI composite index, as shown in Figure 2c.

In order to analyze the spatial change trend of NPP and POI, this study constructs isolines for it and calculates the population center of gravity in Wuhan. With this as the origin and 2° as the interval, 180 section lines were drawn out, and element nodes were generated with 500 m as the interval, which was used to extract the spot values in different directions and distances in Wuhan (Figure 3).

#### 3.2.2. Analysis of the Relationship between NPP and POI and Urban and Rural Spatial Structure

The urban core area refers to the area with the highest development intensity in the city, which provides economic, political, cultural, and other public services and space for the city and its radiation area. The agglomeration effect makes a large number of human activities and service facilities gather here. As a transitional area between the urban core area and rural area, the urban-rural fringe has significantly reduced the intensity of human activities compared with the urban core area and is different from the urban core area in space and function. It can be seen from Figure 3 and Figure 4 that with the increase in the distance from the population center of gravity, the NPP and POI value shows a rapid decline, and the first mutation occurs between the urban core area and the URF.

The urban-rural fringe is closely connected with the city in terms of population, economy, and social activities and should be considered a part of the built-up area. After constructing isolines for the NPP and POI composite index, it is found that the distance between isolines in the urban core area is very close due to the concentration of human activities and various service facilities. The distance between isolines in the URF is relatively increased, and the NPP and POI value fluctuates greatly. In the countryside, the isolines become very sparse, and the NPP and POI value drops rapidly and approaches zero (Figure 4). Further analysis of the relationship between the enclosed area of the NPP and POI contour line and the distance from the population center of gravity shows that, with the increase in the distance from the population center of gravity, the cumulative area value roughly goes through three stages of change, namely, the gentle growth stage of the urban core area, the fluctuating growth stage of the semi-urban area and the rapid improvement stage of the countryside (Figure 4). The second mutation of NPP and POI occurs between the URF and the countryside.

According to the secondary change law of NPP and POI in urban and rural space, this paper takes the area where the value of NPP and POI decreases sharply as the inner boundary of the urban-rural fringe and the area where the isosurface area of NPP and POI increases sharply as the outer boundary of the urban-rural fringe. The spatial scope of the URF can be obtained by connecting the NPP and POI comprehensive index calculated by the breaking point method and the inflection point of the isosurface change.

### 3.3. Delineation of Test Methods for Results

Generally speaking, the typical urban core area and rural land use types are relatively simple, and the information entropy of land use structure is relatively low; The construction land in the urban core area is distributed continuously, and the NDVI is relatively low. The rural area is dominated by agricultural lands, such as farmland and garden land, and ecological land, such as forest land, and the NDVI is relatively high [38]. The urban-rural fringe is a transitional zone between cities and villages. In the process of urban sprawl, its natural underlying surface has been constantly transformed. NDVI is at a medium level and in a downward trend, and the land use structure is highly disordered. According to this rule, this study establishes a 3 km × 3 km grid in the administrative division of Wuhan City, and the verification transects were drawn to compare and analyze the information entropy of land use structure and NDVI spatial characteristics of the above three areas to verify the accuracy of the delineation results, and compared with the population density data. Figure 5 shows the technical process of this study.

## 4. Results

### 4.1. Identification Results of Inner Boundary of Urban-Rural Fringe

After connecting the breakpoints of the NPP and POI composite index in 180 directions, the inner boundary of the urban-rural fringe of Wuhan was obtained. The study found that under the influence of nature, economy, politics, etc., the values of breaking points in different directions are different, but most of them fluctuate around 0.1. In this study, 0.1 is used as the threshold value of the inner boundary of the URF to correct the abnormal breakpoints in individual directions (Table 2, Figure 4). On the whole, Wuhan’s NPP and POI is higher in the middle and lower around. The urban core area has the highest value, fluctuating between 0.2 and 0.6, while in the URF and rural areas, the value drops to below 0.1, and in rivers and lakes, NPP and POI drops sharply to 0. At the same time, there are also seven cluster-like areas with high NPP and POI values between 0.1 and 0.3 outside the major urban core area. After comparing and analyzing the spatial location of these seven discrete high-value areas with the “Overall Planning of Land and Space of Wuhan City (2021–2035)”, we found that they match with the sub-city and new city groups in the planning. In this study, they are uniformly identified as new city clusters. The results show that the main urban core areas in Wuhan mainly include Jiang’an District, Jianghan District, Qiaokou District, Hanyang District, Wuchang District, Qingshan District, and the west of Hongshan District. The new city clusters are located in Songjiagang, Yangluo, Caidian, Zhifang, Shamao, Qianchuan, and Zhucheng, respectively (Figure 6).

### 4.2. Delineation of Spatial Scope of Urban-Rural Fringe

After connecting the inflexion points of the accumulated area values of NPP and POI isolines obtained by the breakpoint algorithm, the outer boundary of the urban-rural fringe was obtained. After combining the inner and outer boundaries, the space enclosed between the two boundaries is the range of the URF of Wuhan City (Figure 6).

## 5. Discussion

### 5.1. Verification of Urban-Rural Fringe Identification Results

#### 5.1.1. Overall Space Measurement

After calculating the population density, land use structure information entropy, and NDVI of Wuhan’s urban core area, urban-rural fringe, and rural areas in 2020, respectively, as defined in the previous text, the population density in the urban core area is 9335.83 persons/km^2^, 2556.28 persons/km^2^ in the URF, 827.84 persons/km^2^ in rural. The population density presents the law of decreasing from urban core area to URF to rural. The information entropy of land use structure is 0.937 in the urban core area, 1.630 in the URF, and 1.418 in the rural, which conforms to the characteristics of higher in the URF and lower in the urban core area and rural area. The NDVI in the urban core area is 0.431, its standard deviation is 0.101, the URF is 0.523, its standard deviation is 0.184, the rural area is 0.718, and its standard deviation is 0.158. The NDVI shows the characteristics of increasing from urban core area → URF → rural. The NDVI standard deviation shows the characteristics of higher in the URF and lower in the urban core area and rural area.

The NDVI and its standard deviation measurement results of Wuhan from 2010 to 2020 in Figure 7 show that the NDVI in urban core areas and rural areas shows an increasing trend with a low standard deviation, which is closely related to urban green infrastructure construction, rural land remediation, and other activities. The NDVI in the URF shows a decreasing trend, which is related to the change in the nature of the underlying surface caused by urban space expansion. Its standard deviation is high, which conforms to the distribution law of higher in the URF and lower in the urban core area and rural area (Figure 7). In Figure 7, AVG stands for average value of NDVI, SD stands for standard deviation of NDVI, UCA stands for urban core area, URF stands for urban-rural fringe, and RA stands for rural area.

#### 5.1.2. Transects Verification

In order to further verify the accuracy of the delineation results, this study established a 2 km × 2 km verification grid at first to calculate the land use structure information entropy and NDVI values in the verification unit. However, due to the large patch of land use, this method cannot well reflect the difference of land use structure information entropy between grids, but when the verification unit expanded to 3 km × 3 km, the sensitivity of information entropy of land use structure is significantly improved. Therefore, this study established a 3 km × 3 km verification grid in the administrative division of Wuhan City, derived four verification transects: north–south, west–east, northwest–southeast, northeast–southwest, and numbered the validation units in each validation transect from the number 1 to calculate the land use structure information entropy and NDVI values in the verification unit (Figure 8). It can be seen from Figure 8 that the information entropy of land use structure in Wuhan can be divided into the low-value area of 0.00~0.70, the middle-value area of 0.70~1.20, the high-value area of 1.20~1.76 by natural breakpoint method, and NDVI can be divided into the low-value area of 0.13~0.54, the middle-value area of 0.54~0.70, and the high-value area of 0.70~0.92 by natural breakpoint method. The accuracy of the delineation results can be tested by examining the land use structure information entropy and NDVI numerical value level in the sample belt and analyzing its spatial change law.

Figure 9a–d shows the calculation results of the land use structure information entropy and NDVI of the four validation transects, where the horizontal axis represents the validation unit number in each direction, and the blue background area represents the URF. The numbers in Figure 9e represent the validation unit number in each validation transect, and the image in Figure 9e is the remote sensing image of example validation units. The results show that the information entropy of land use structure of the 20th–25th and 33rd–38th verification units in the north-south transect, the 4th–5th and 18th–22nd verification units in the west-east transect, the 4th and 11th–13th verification units in the northwest-southeast transect, and the 7th–9th and 18th–19th verification units in the northeast-southwest transect is generally greater than 1.2, belonging to the high-value area, their NDVI fluctuates between 0.5 and 0.7, belonging to the median area, which is a typical characteristic of the semi-urban area (Figure 9). Among them, the information entropy of land use structure of verification units of 1st–19th and 39th in the north–south transect has fluctuated, and the value in some areas is greater than 1.2, which is related to a large number of interlaced distribution of cultivated land, garden land, forest land, and grassland, and their NDVI is greater than 0.6, vegetation coverage is high. The information entropy of the land use structure of the 40th verification unit is higher than 1.2, and its NDVI is at the median level, between 0.5 and 0.6. This is because the mining land of this unit is distributed contiguously and interspersed with cultivated land, garden land, forest land, and other types of land, but almost no residential and commercial land is distributed, living service facilities are extremely scarce, NPP and POI values are low, and they are still rural areas (Figure 9a,e). In the west–east transect (Figure 9b,e), the 8th verification unit is affected by the distribution of lakes, and the information entropy of land use structure increased slightly; The 25th verification unit is located in the distribution area of Zhangdu Lake, the NDVI and information entropy of land use structure decreased sharply.

The calculation results show that, among the four verification transects, the areas where the land use structure information entropy is generally greater than 1.2 and the NDVI between 0.5 and 0.7 is consistent with the range of the URF defined above. The areas with land use information entropy less than 1.3 and NDVI less than 0.5 and greater than 0.7 are consistent with the urban core area and rural area defined above; they conform to the law of gradual decrease in land use structure information entropy from the URF → rural → urban core area, and the law of gradual increase in NDVI from rural → URF → urban core area.

#### 5.1.3. Field Verification

In order to comprehensively assess the authenticity of the delineation results of the urban-rural fringe, this paper selects some typical areas for field verification and analysis (Figure 10). The a–i in Figure 10 shows the distribution of points in the field survey.

Figure 10a,b are respectively located on Jianghan Road and Han Commercial Pedestrian Street. The survey shows that these areas are full of high-rise buildings, active businesses, high floor area ratio and population density, high road hardening rate, sparse vegetation, and there is light performance at night in addition to necessary lighting, which is consistent with the urban core area identified in the text. Figure 10c,d are respectively located in Qingfeng Village, Huangpi District, and Dongfang Village, Jiangxia District. Buildings in these areas are low and old and sparsely arranged along the roads, commercial and public service facilities are extremely scarce, population density is low, the farmland is distributed continuously with high utilization rate, and the vegetation coverage rate is high, presenting typical rural landscape characteristics.

The area in Figure 10e belongs to the Yangtze New City. Since the official approval in 2022, a large number of construction projects have settled here. The scale of land transfer is large. The commercial service facilities are mainly built around the newly built residential areas, providing daily services such as catering, express delivery, and car maintenance for residents in the nearby area, the number and scale of them is relatively small. Figure 10f is located on the Shidao Line in Huangpi District. The Shekou Central Primary School beside the road was renovated in 2018. There is a storage market in the west, some self-built houses in the east, and small farmlands on the north side. There are few commercial and service facilities along the road. The convenience store on the roadside has the function of integrating business, residence, and living. Jialing Village in Figure 10g is located on the west bank of Yanxi Lake. In order to speed up the construction of public health service facilities, this area has been allocated to Wudong Hospital, part of the demolition work has been completed, and the construction of the mental health rehabilitation center in the west has started. The villagers in the east have left small vegetable plots and dilapidated houses after relocation. Figure 10h is located in Biolake. According to the survey, there are removal settlement houses in CSCEC Xingguang Community under construction. To the south is the construction plot of Phase II of the Biological Innovation Park. This area is planned to be built into a new biological industry city integrating research and development, production, logistics, and living. A large number of residential bases and agricultural land have been converted into construction land, and related supporting facilities are in the development and construction stage. Figure 10i is located in Xingyuanchang Village, Daqiao New District, where a series of industrial parks and high-rise buildings have been built. However, there are still small abandoned vegetable gardens in the area to be demolished and constructed, with garbage everywhere and weeds overgrown. Some villagers still retain their traditional farming life. The field survey shows that the landscape of the above areas is characterized by the staggered distribution of vegetable gardens, industrial parks, low-density self-built houses, high-density residential quarters, new commercial and public service facilities, and land under construction. The land use types of them are diverse, the population and vegetation density are between the urban core area and the rural area, and they are in the transition stage from rural to urban, matching the urban-rural fringe defined in the text.

### 5.2. Spatial Analysis of Urban and Rural Fringe in Wuhan

It is found in the calculation that large lakes have a significant impact on the direction of urban expansion in Wuhan. Taking section lines 165–180 as an example, NPP and POI fluctuates between 0.15 and 0.4 within 0–6 km from the population center of gravity and drops sharply to 0 after exceeding 6 km. In the west of section lines 89–96, NPP and POI remained at a high value of about 0.5 at about 6 km away from the population center of gravity and declined to about 0.1 at 15–20 km. The former is obstructed by the East Lake, which makes it difficult to continuously promote urban construction and sharply reduces human activities. The latter is the distribution area of Jianghan District and Qiaokou District (Figure 11) because of the continuous plains, fewer water barriers, early urbanization, and high level. This kind of situation is also shown in large lake areas such as Houguan Lake, Zhiyin Lake, Tangxun Lake, Jinyin Lake, etc. The inner boundary of the URF protrudes toward the urban core area, while in the lake areas such as Majia Lake, Baishui Lake, Yandong Lake, Baoxiehou Lake, Niushan Lake, Lu Lake, Guanlian Lake, Zhushan Lake, etc., the outer boundary of the URF is concave inward (Figure 6).

The urban-rural fringe area of Wuhan is 1482.35 km^2^ in total, and it accounts for 17.30% of the city’s total area. This result is slightly smaller than the 1746 km^2^ area of the URF of Wuhan in 2016 identified by Xiong, N. using NPP-VIIRS as the data source, which may be related to the saturation and spillover effect of NPP-VIIRS [40]. However, the area ratio of URF determined in this study is very similar to that determined by Zhang, J. using NTL and POI fusion data. Zhang, J. et al. fused NTL and POI with wavelet transform and identified that the area of URF in Kunming in 2020 accounted for 17.37% of its total area [32]. At the same time, Lei, Z.C. used POI to identify the URF of the Guangzhou-Foshan Metropolitan Area, and the result showed that its URF accounted for 19.18% of the total area of the Guangzhou-Foshan Metropolitan Area, which was similar to the result of this study. The land use types of URF in Wuhan are mainly construction land, water area, and cultivated land, accounting for 40.75%, 30.03%, and 14.60% of the total area of URF, respectively (Figure 12 and Table 3).

Under the evolution of the history of three towns and the “gene” of big rivers and mountains, the urban-rural fringe of Wuhan City, affected by transportation, economy, and policies, presents a “six axes, two rings” banded and leaping distribution characteristics around the main urban core area of Wuhan. The “six axes” are Yangluo—Shamao Yangtze River Expansion Axis, Zoumaling—Caidian Han River Expansion Axis, Baoxie—Zuoling to Ezhou Expansion Axis, Songjiagang—Boquan to Xiaogan Expansion Axis, Zhifang to Xianning expansion axis, Dunyang—Changfu to Xiantao Expansion Axis, and the “two rings” refer to Qianchuan—Sheshui Jumping Expansion Ring and Zhucheng—Jushui Jumping Expansion Ring. It can be seen that the distribution and trend of large rivers, as well as the deep communication and integration of the Wuhan City circle, have a profound impact on the spatial morphology of the URF of Wuhan (Figure 13).

### 5.3. Comparative Analysis

Comparing the delineation results with the distribution of the population density of WorldPop, it is found that the scope of the URF of Wuhan is consistent with the general trend of the population density distribution, but there are some differences. The specific performance is that its inner and outer boundaries have expanded outwards relative to the high and median population density area. Figure 14 shows the spatial distribution of population density and the spatial location of the URF in Wuhan, where a–d shows the remote sensing images of sample areas. It can be seen from Figure 14a,b that the population density of verification units a and b is between 0 and 83.86 persons/hm^2^, which belongs to the low-value area, but the image shows that this type of verification unit has rich landscape types, with forest land, cultivated land, waters, buildings under construction and existing buildings staggered. After comparing it with the Master Plan of Land and Space of Wuhan City (2021–2035), it is found that the verification unit a is the radiation range of the Songjiagang–Hengdian development axis, and the verification unit b is the radiation range of the Sino French Eco city–Caidian development axis. It is in the period of vigorous development and construction, that is, the transition from rural to urban. It is difficult to reflect the development and utilization of such transitional areas in a timely manner only by population density. The c verification unit is located in Jingang New Area. Since its approval at the end of 2011, a large-scale industrial park has been built. Although the population density is low, its demographic composition, industrial type, and landscape characteristics are no longer rural. The d verification unit is located on Canglong Island. Although its population density belongs to the median area, it was developed early and has built an ecological new town integrating technology, ecology, residence, tourism, and other functions. It does not belong to the URF. It can be seen from the above that when the population density is in the low-value range, NPP and POI can accurately distinguish the differences between rural areas, development and construction zones, and industrial parks. When the population density is in the median range, it can distinguish the differences between the URF and ecological new cities. The identification results of NPP and POI are more accurate than the population density distribution data.

Land use is a relatively traditional method in the division of the urban-rural fringe, and its feasibility has been confirmed by a large number of studies. However, when using the information entropy of land use structure to delimit the URF, in order to ensure the sensitivity of the information entropy, a larger grid is usually used for division. For example, the accuracy of the URF divided by Qian et al. using the information entropy of land use structure was 1 km, which led to low accuracy of the delineation results [41]. As one of the important parameters reflecting crop growth and nutrition information, NDVI has stronger natural attributes and can only reflect human activities laterally. Therefore, land use and NDVI are used to test the accuracy of demarcation results. When nighttime light data are applied to the spatial delimitation of the URF alone, the accuracy of the delimitation result is relatively low due to the “saturation”, “overflow,” and resolution of the image. Zeng, T.Y. et al. used DMSP-OLS and NPP-VIIRS nighttime light data to conduct long-term time series monitoring of the URF in Shenyang from 2013 to 2020. It was also found that the nighttime light data had low resolution and was difficult to display small-scale rural lights., so they suggested combining it with other high-resolution data such as Landsat 8 and SPOT [2]. As a kind of point set data, POI usually needs to be processed to simulate the spatial change trend when describing the urban spatial characteristics. At present, few studies have used POI alone to divide the URF, mainly because it is difficult to reflect the scale and development intensity of the region and its industries. However, some studies have confirmed that the fusion data of NTL and POI can make up for the “hole” phenomenon caused by low lighting in cities; this is mainly because POI data, as a collection of geospatial quantities, can make up for the lack of observation of NTL at the micro scale [42]. This is basically consistent with the conclusion of this study, that is, using NPP and POI to identify the URF is more accurate than using them alone. At the same time, this study has identified seven new city clusters with NPP and POI between 0.1 and 0.3, which cannot be achieved when using POI alone, indicating that the application of NPP and POI in the division of urban-rural fringe can better represent the vitality of urban development compared with using POI alone.

### 5.4. Limitations

Although NPP and POI can distinguish natural landscape from man-made landscape, its internal value in agricultural land, ecological land, and other natural areas is extremely low, which is difficult to reflect its internal spatial structure and accurately characterize its human activities. The spatial subordination of such areas needs to be further explored. At the same time, the uncertainty and incompleteness of NTL and POI as open-source data are also worth considering.

This paper only takes Wuhan as an example to discuss the urban and rural spatial change rules of NPP and POI and delimit the spatial scope of its urban-rural fringe. It does not compare and analyze cities with different areas, populations, and economic development conditions. The applicability of this index to the areas demarcated in the urban-rural fringe remains to be discussed.

The boundary of the URF is actually changing with the development of the city. This study only demarcates the boundary based on a one-time point, which is relatively weak in guiding urban and rural planning. In the future, if the scope of the URF can be continuously simulated and predicted, it is of great significance to analyze the speed and direction of urban expansion and assist planners in better spatial planning.

## 6. Conclusions

Accurately delineating the spatial scope of the urban-rural fringe is of great significance for recognizing the urban and rural spatial structure, controlling the urban sprawl, and promoting the integrated development of urban and rural areas. It coincides with the planning concepts of breaking down barriers, coordinating urban and rural areas, and ecological construction in the global new round of land and space planning. Aiming at the problems of single data source selection, difficulty in obtaining data, and low spatial and temporal resolution of recognition results in the past, this study applies POI and Nighttime Light fusion data to the spatial delineation of urban-rural fringe, and the recognition results have been greatly improved in objectivity, timeliness and accuracy compared with using NTL or POI alone. The main conclusions are as follows:(1)NPP and POI composite index is applied to the study of spatial delineation of urban-rural fringe, which can not only give full play to the attributes and micro-advantage potential of POI, synthesize the difference between the type of facilities and light intensity, reduce the “saturation” and “overflow” effects of Nighttime Light, but also master the direction of urban and rural development in a large scale by using NTL, reduce broken and isolated spots, and ensure the continuity of identification results. Compared with using POI, NTL, or population density data alone, the accuracy is higher, and the timeliness is more efficient.(2)NPP and POI can quantitatively identify potential central area and multi-layer structure of the city. The urban-rural fringe area of Wuhan is 1482.35 km^2^, which accounts for 17.30% of the total area of Wuhan City. Around the main body of the urban core area, it presents the “six axes and two rings” banded and leaping distribution characteristics. The urban core area of Wuhan City is in the form of “one main area with multiple cores”, and the rural areas are widely distributed in the periphery of Wuhan City, presenting a continuous distribution feature. This characteristic shows that NPP and POI can better represent the vitality of urban development than population statistics, land use, and landscape. The identification results have important reference significance for the allocation of global urban infrastructure, industrial division, ecological function division, and other studies.(3)NPP and POI composite index shows a double mutation law in urban and rural spaces. The first mutation occurred between the urban core area and the URF. With the increasing distance from the urban core area, NPP and POI showed a rapid downward trend. The second mutation occurred between the URF and the rural. With the increasing distance from the URF, the cumulative value of NPP and POI isosurface area showed a sharp upward trend. This law confirms that the urban-rural fringe, as a regional entity generated in the process of urban expansion, objectively exists, has spatial continuity, transition of factors such as economy, population, and land, and consistency of internal characteristics, which is of empirical significance for the discussion of the ternary and binary models in the study of cities spatial structure.(4)This paper only takes Wuhan City as an example to discuss the urban and rural spatial change law of NPP and POI and delimit the spatial scope of its urban-rural fringe. After the data are enriched in the future, cities with different areas, populations, and economic development statuses can be compared and analyzed to discuss the universality. Although NPP and POI can distinguish between natural landscape and man-made landscape, its value is extremely low in agricultural land, ecological land, and other natural areas, and the spatial subordination of such areas needs to be further discussed. NPP and POI can represent the vitality of development, with convenient access and a short renewal cycle. If machine learning and other methods can be used to build the spatial range of URF at different time points, it is of great significance to analyze the speed and direction of global urban expansion, urban shrinkage, and other issues.

## Figures and Tables

**Figure 1 ijerph-20-04395-f001:**
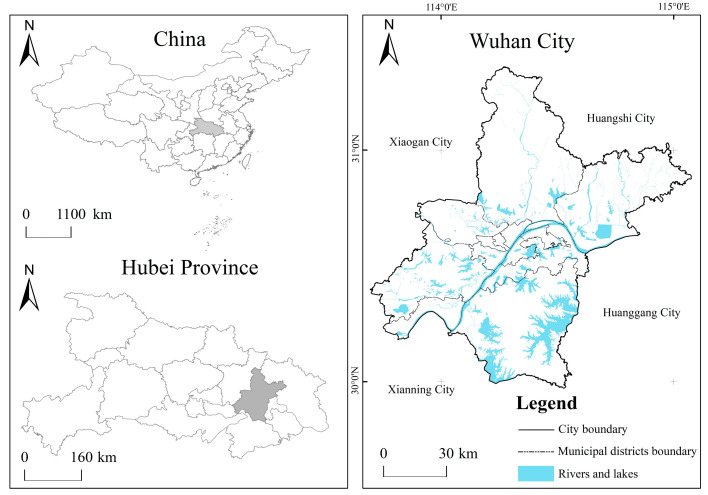
Location Map of Wuhan City.

**Figure 2 ijerph-20-04395-f002:**
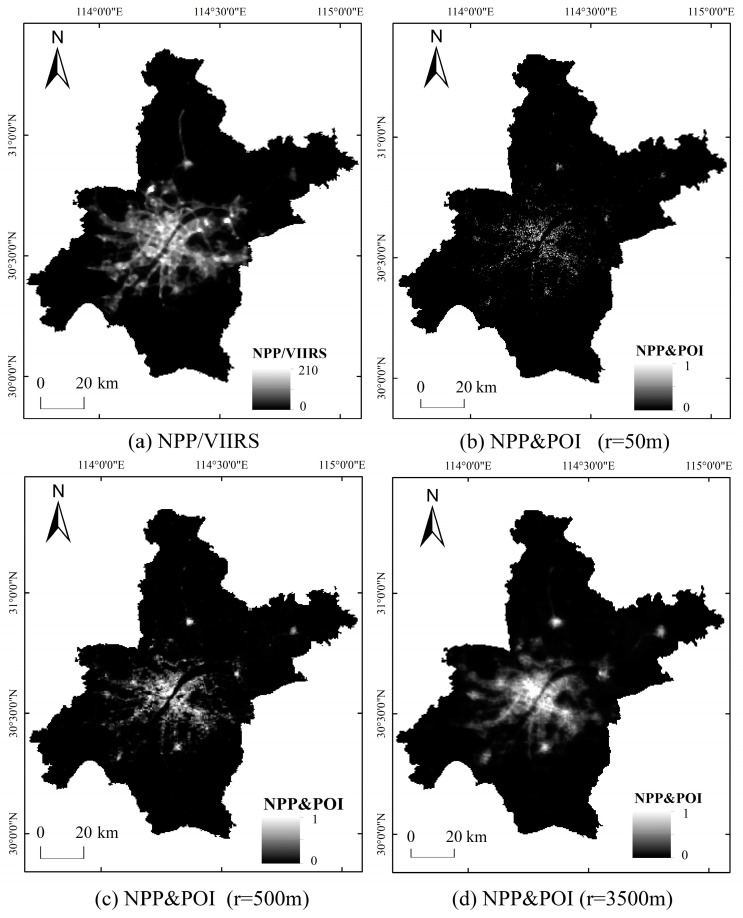
Comparison of NPP and POI processing effects in Wuhan under different search distances.

**Figure 3 ijerph-20-04395-f003:**
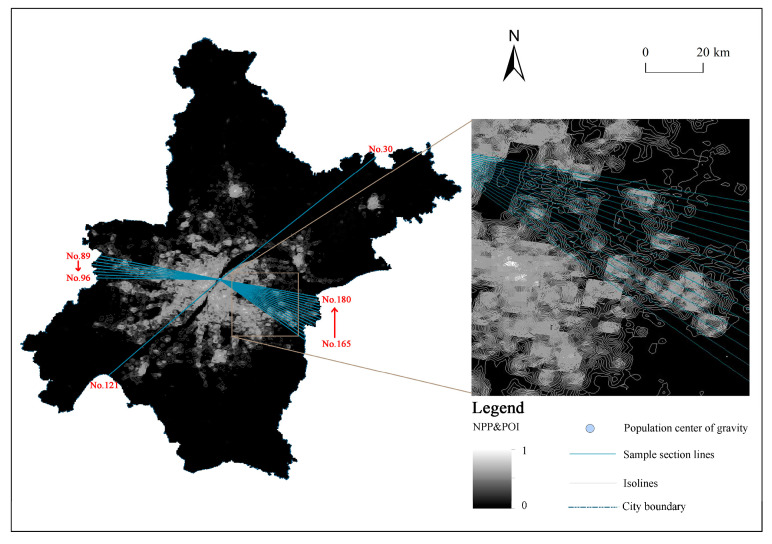
Schematic diagram of NPP and POI section lines and isolines.

**Figure 4 ijerph-20-04395-f004:**
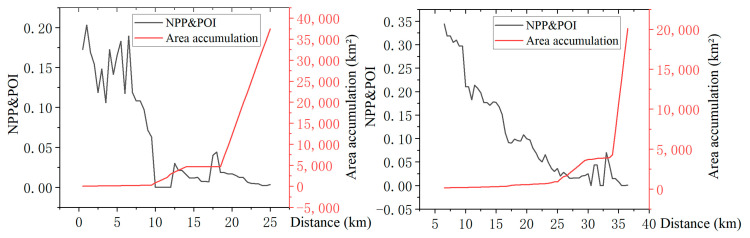
Variation of NPP and POI and its isosurface area with distance (take 30 and 121 transects as an example).

**Figure 5 ijerph-20-04395-f005:**
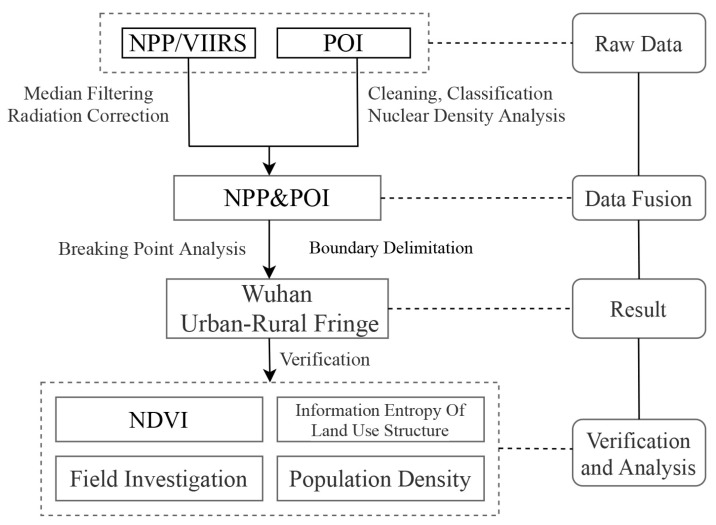
Technical flow chart of delimitation of URF in Wuhan.

**Figure 6 ijerph-20-04395-f006:**
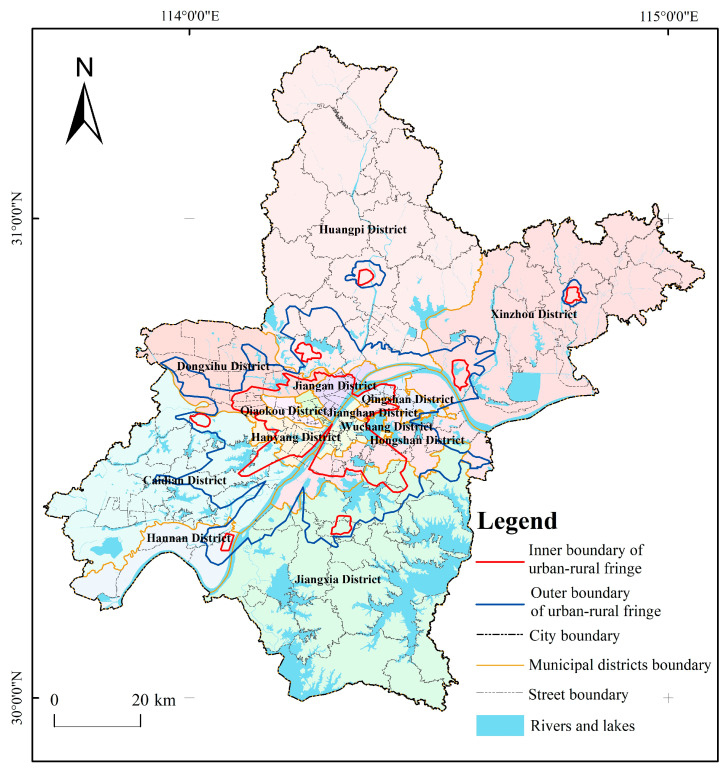
Wuhan urban-rural fringe boundary.

**Figure 7 ijerph-20-04395-f007:**
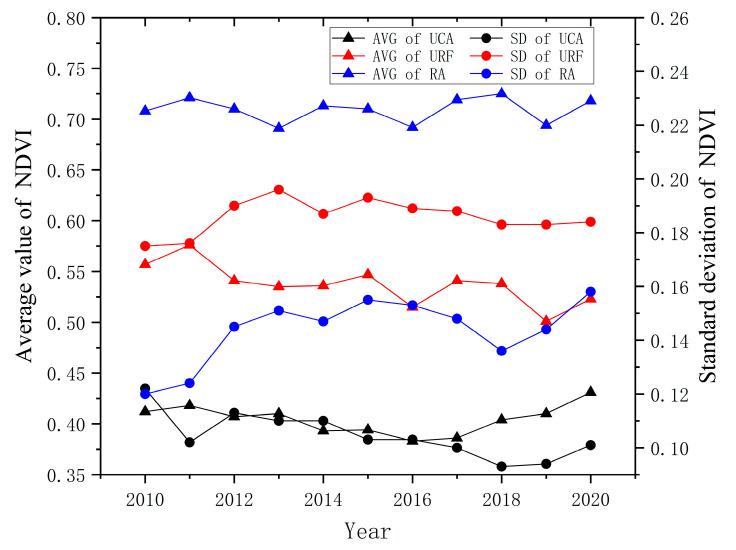
NDVI calculation chart of Wuhan urban core areas, urban-rural fringe, and rural areas from 2010 to 2020.

**Figure 8 ijerph-20-04395-f008:**
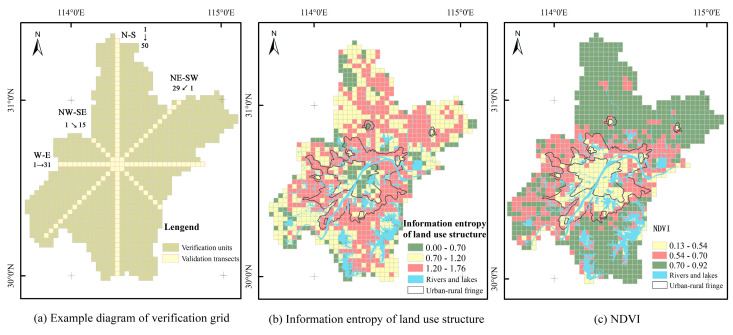
Verification grid and its measured value.

**Figure 9 ijerph-20-04395-f009:**
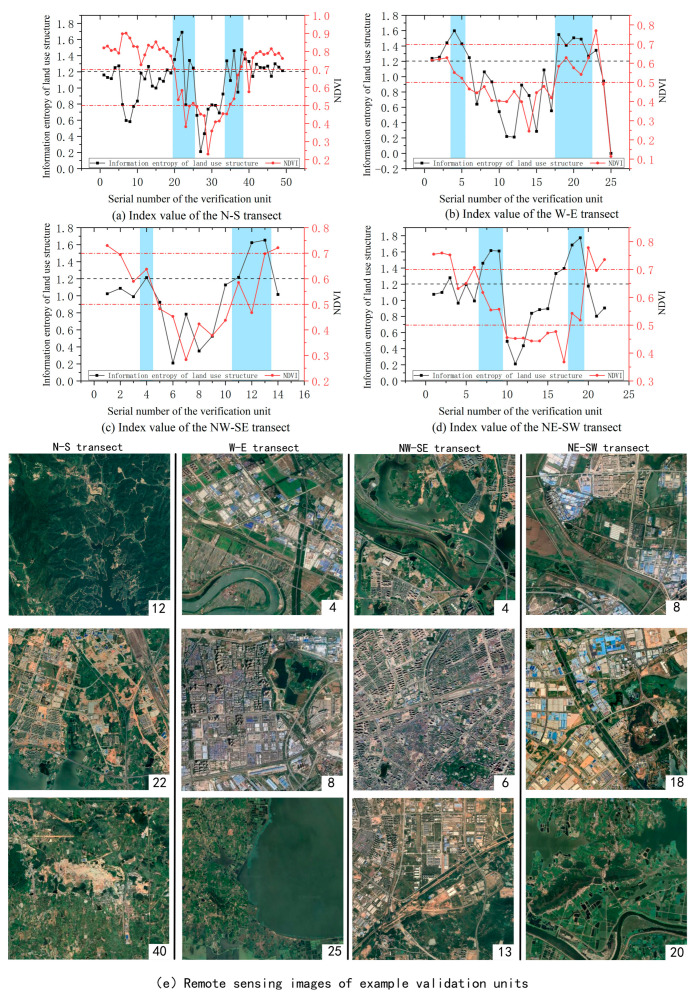
Analysis results of Verification transects.

**Figure 10 ijerph-20-04395-f010:**
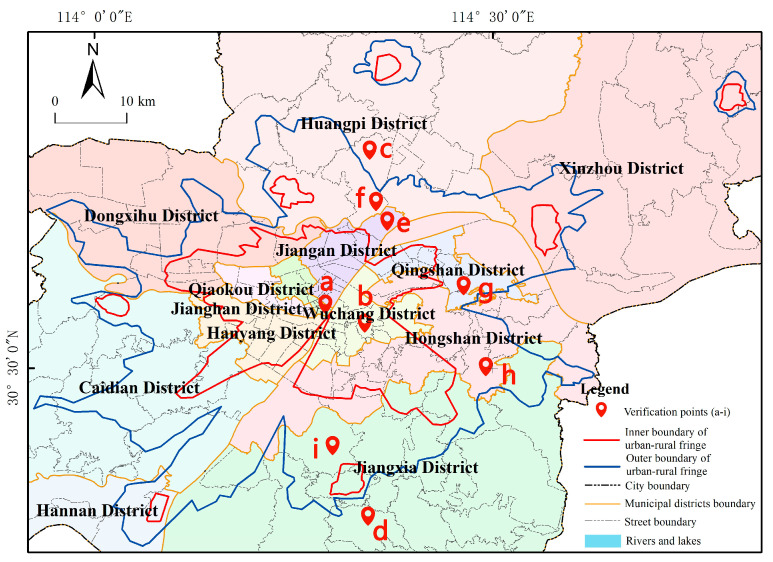
Distribution Map of Site Investigation Points Wuhan City.

**Figure 11 ijerph-20-04395-f011:**
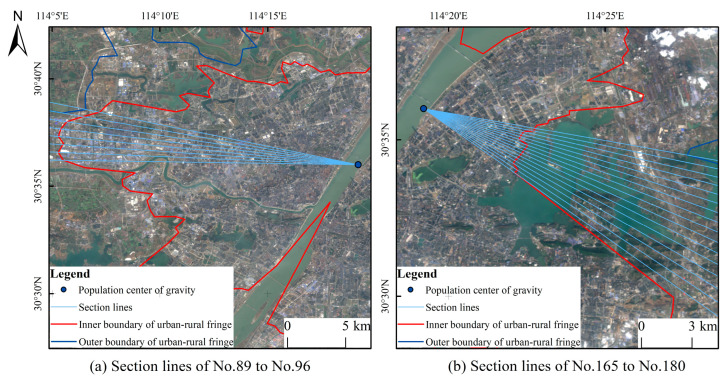
Schematic diagram of remote sensing image at section lines.

**Figure 12 ijerph-20-04395-f012:**
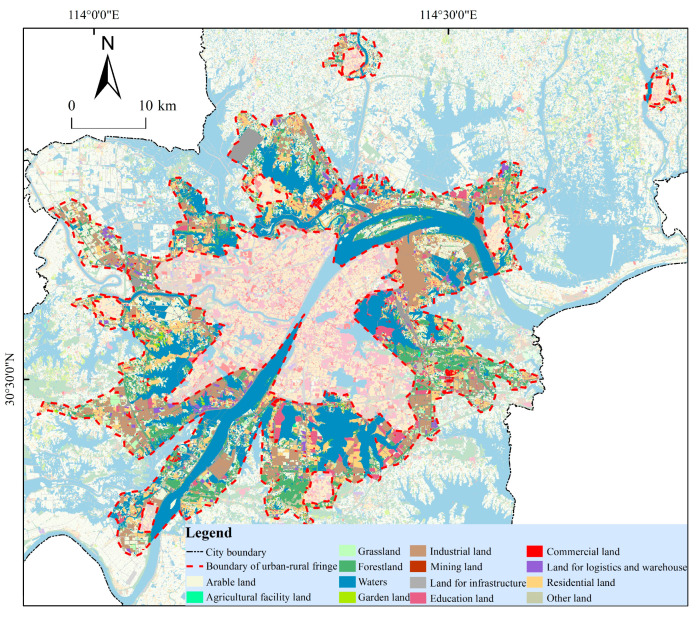
Distribution Map of Land Use Types in the Urban-Rural Fringe of Wuhan City.

**Figure 13 ijerph-20-04395-f013:**
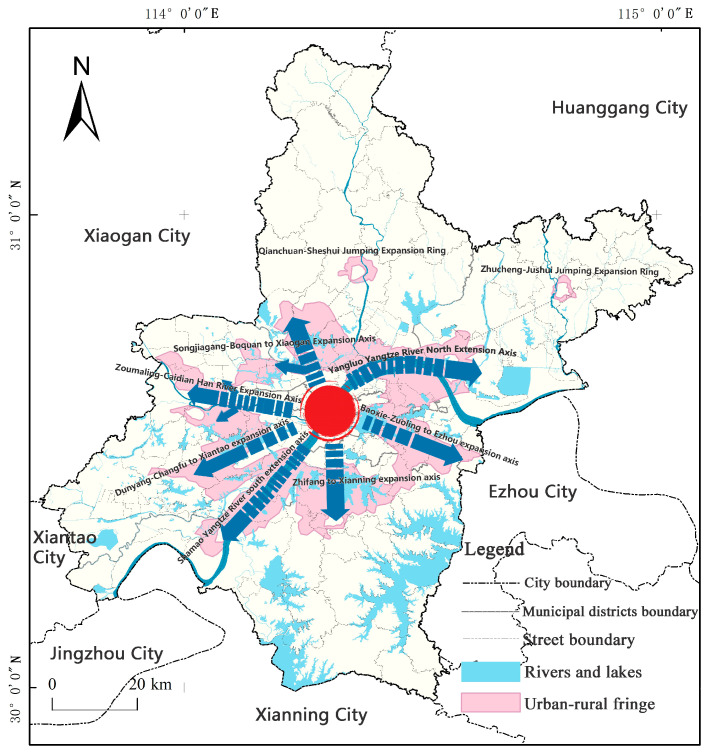
Spatial form of Wuhan Urban-Rural Fringe.

**Figure 14 ijerph-20-04395-f014:**
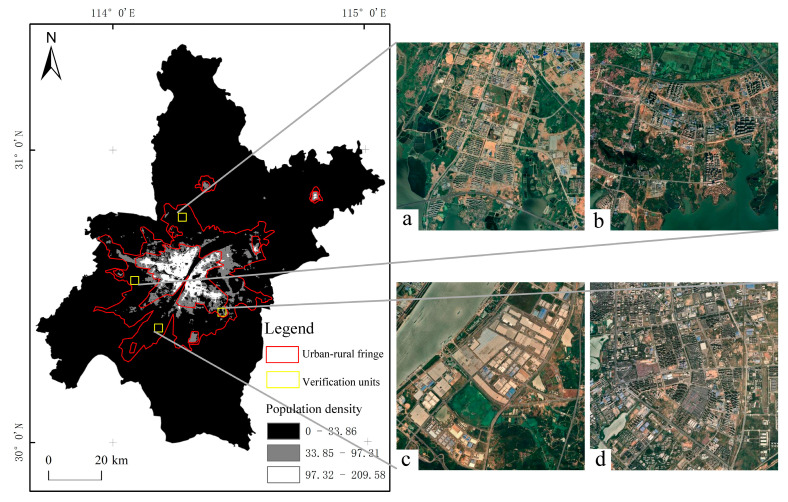
Distribution Map of Boundary and Population Density in the Urban-Rural Fringe of Wuhan City.

**Table 1 ijerph-20-04395-t001:** Classification Statistics of POI in Wuhan.

POI Function Category	POI First Level Industry Classification in Baidu Map	Number of POIs
Cultural and Sports	Tourist attractions, leisure and entertainment, sports and fitness, education and training, cultural media, natural features	53,966
Commercial	Food, hotel, shopping, life service, beauty ^1^, car service, finance	68,508
Industrial	Corporate	70,204
Public service	Medical treatment, transportation facilities, government agencies	33,640
Residential	Real estate	119,543

^1^ Note: “Beauty” refers to commercial service facilities such as beauty, hair, nail, and body.

**Table 2 ijerph-20-04395-t002:** Statistics of NPP and POI Breakpoint Values on Different Profile Lines.

Order Number of Section Lines	Average Value of Breaking Point of NPP and POI Composite Index
1~30	0.109
31~60	0.095
61~90	0.105
91~120	0.103
121~150	0.097
151~180	0.114

**Table 3 ijerph-20-04395-t003:** Area Statistics of Land Use Types in the Urban-Rural Fringe of Wuhan City.

Land Category Name		Area/km^2^	Proportion/%
Agricultural land	Cultivated land	216.42	14.60
	Agricultural facility land	1.49	0.10
	Sum	217.91	14.70
Ecological land	Grassland	34.51	2.33
	Forestland	161.72	10.91
	Waters	445.15	30.03
	Garden land	5.58	0.37
	Sum	646.96	43.64
Construction land	Industrial land	168.55	11.37
	Mining land	5.41	0.36
	Land for infrastructure	175.11	11.81
	Education land	52.44	3.54
	Commercial land	20.54	1.39
	Land for logistics and warehouse	20.4	1.38
	Residential land	161.61	10.90
	Sum	604.06	40.75
Other lands	Other land	13.42	0.91
	Sum	13.42	0.91
Total		1482.35	100

## Data Availability

All data collected during the study are available in the submitted article and the detailed data values are available from the corresponding author by request.

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
