# Peer review of "Research on Spatial Delineation Method of Urban-Rural Fringe Combining POI and Nighttime Light Data—Taking Wuhan City as an Example"

_ijerph, 2023, doi:10.3390/ijerph20054395_

Round 1

Reviewer 1 Report

This study proposed a composite index fusing NTL and POI data for urban-rural fringe (URF) delineation. However, after reading the manuscript, I have several concerns regarding methodology and research design. As a result, I have to decline it at the current stage.

Major concerns:

1. For the verification parts, the analyses are mostly qualitative descriptions and lack comparisons. The authors need to prove why the combination of NTL and POI data is more accurate in delineating URF compared with using them alone and how the proposed method is superior to other state-of-arts methods.

2. Separate the results and discussions into two sections. Present verification results (sections 4.5-4.6) first, and then other analytic results. Besides, summarize the pros and cons of the proposed method in the Discussion section.

3. Add appropriate descriptions to the titles of all figures. Currently, the descriptions are quite simple and incomprehensible. For example, in Figure 10(e), what do numbers in the lower-left stand for?

Minor issues:

Line 29-31. I did not see the delineation results and comparisons of using these data alone in the main text and was not convinced of “more accurate and time-sensitive”.

Line 31. NPP? Write the full name where the abbreviation first appears.

Lines 114-116. As defined here, the sharp drop in population is an essential characteristic of URF. So why not incorporate the population distribution data like WorldPop or GPWv4 in the methodology?

Line 179. Add a figure to show the overall flowchart of the designed method.

Line 199. Converes?

Line 226. Why set bandwidth as 50, 500, and 3500 m? Any criteria for parameter setting?

Line 228. What does value (0-1) represent in Figure 2(b-d)?

Line 251. The Chinese character “号” should not appear in Figure 3.

Lines 308-309. “they match with their sub cities and new city clusters, which are identified as new city clusters in this study”. What does the sentence mean?

Lines 323-339. Not necessary to introduce so many details (xxx street) here.

Line 342. Have you considered the impact of large lakes on the delineation of URF? It seems that a universal threshold is used in the study.

Lines 359-362. Does 1482.35km² mean an overestimation or underestimation? Have you compared your results with other datasets or official statistics?

Line 402. Table 4. Maybe a line chart is more suitable for showing the trend of NDVI.

Line 407. Why 3km * 3km?

Line 419. Figure 9. Overlap the delineation results with land use structure information entropy and NDVI.

Lines 420-424. Better add a figure to show the spatial correspondence between the number and direction (e.g., 11th-13rd verification units correspond to northwest-southeast transect).

Lines 443-450. Why not consider incorporating land use, NDVI, or population data for identifying URF?

Lines 534-535. The spatial and temporal resolution of recognition results is not high as well.

Line 537. From the present results, I cannot see how the timeliness and accuracy of the recognition results are improved.

Author Response

Please see the attachment, thank you!

Reviewer 2 Report

It remains to explain the different units of measurement used by each source of information. For example, NPP&POI what kind of units are you mixing? . In line 320, you don´t explain how you get the breakpoint algorithm.

The introduction should be modified. The point 2, you expose some units which are not standard units so that's not very understandable.

Author Response

Please see the attachment, thank you!

Reviewer 3 Report

Thank you for giving me this opportunity to read the manuscript entitled "Research on Spatial Delineation Method of Urban-Rural Fringe Combining POI and Nighttime Light Data -- Taking Wuhan City as an Example". Overall, the study presents a promising approach for identifying the urban-rural fringe (URF) by combining Point of Interest (POI) and Nighttime Light (NTL) data. The findings of the study indicate that the fusion of POI and NTL data can provide more accurate and time-sensitive URF boundaries compared to those obtained from POI, NTL, or population density data alone.

However, the study has some limitations that need to be addressed before publication. The main limitations include:

1.     Please replace the keywords that already appear in the manuscript's title with close synonyms or other keywords, which will also facilitate your paper being searched by potential readers.

2.     The study could benefit from a comparison with existing methods for URF delineation to further demonstrate the advantages of the proposed approach.

3.     A sensitivity analysis of the results to different input parameters would provide additional confidence in the robustness of the approach.

4.     The study only focuses on Wuhan City and the results may not be generalizable to other cities with different urban and rural structures, which should be treat as a limiatation.

5.     Line 91, “urbanization dynamics estimation [18]…”: a paper titled “How does urban expansion impact people’s exposure to green environments? A comparative study of 290 Chinese cities” is suggested to be added as a reference to support the statement here. Line 102, “extraction of built-up areas [24], …” a paper titled “Dynamic assessments of population exposure to urban greenspace using multi-source big data” is also suggested to be added.

6.     Limitation section should be added as a sub-section to the Discussion.

7.      Some grammatical errors exist in the manuscript. Therefore, a critical review of the manuscript's language will improve its readability.

Author Response

Please see the attachment, thank you!

Round 2

Reviewer 1 Report

I appreciate the authors' considerable effort in improving the manuscript and I recommend that this paper be published.